# Father Involvement and Maternal Stress: The Mediating Role of Coparenting

**DOI:** 10.3390/ijerph20085457

**Published:** 2023-04-10

**Authors:** Dora d’Orsi, Manuela Veríssimo, Eva Diniz

**Affiliations:** William James Center for Research, Ispa—Instituto Universitário, 1149-041 Lisboa, Portugal

**Keywords:** maternal stress, father involvement, coparenting, parenting

## Abstract

In recent years mothers have been finding themselves overwhelmed by the need to balance work and maternal duties. Father involvement in childcare has been related to a decrease in mother’s burden in childcare. This association is influenced by multiple aspects, such as the way parents share parenting perspectives and views about child rearing, i.e., coparenting. Nevertheless, the mediating role of coparenting on the association between father involvement and maternal stress has been overlooked. This will be addressed by the current study. A total of 254 Portuguese married/cohabiting mothers of preschool children reported on maternal stress, father involvement in caregiving, and coparenting. Data was collected in public and private schools through questionnaires and online through advertisements in social media. Results show that greater father involvement in direct care was associated with greater maternal stress, but this direction changes when mediated by cooperative coparenting. Moreover, results suggest that when mothers perceived less conflict in coparenting, greater father (in)direct care contributed to decreased maternal stress. The current study supports the notion that fathers involvement and parent’s cooperation account to mothers’ wellbeing, which will improve family dynamics.

## 1. Introduction

Social, economic, and demographic changes during the last decades introduced significant variations in family dynamics and on parental roles. Women’s increased enrollment in the workforce is one of the most fundamental aspects of this shift, affecting the traditional division of domestic work and parents’ commitment to childcare. Mothers are no longer seen as the only ones responsible for childcare and wellbeing, and father’s involvement is now expected to include child’s caregiving, rather than solely being the family’s breadwinner and the moral guide [1,2,3]. Nevertheless, each parent’s involvement in the child and the household activities tends to remain unequal, with responsibilities related to childcare and domestic chores still mainly performed by women [4,5,6,7]. Hence, mothers face the double burden of work, caregiving, and household chores, which may account for their perception of greater stress, e.g., [3,8]. This is particularly relevant in Portugal, which is not only the European Union (EU) country with more women working outside the home in full-time jobs, but also one of the countries with the lowest rates of gender division of household chores. Portuguese women spend 303 min daily on household chores, whereas men spend 77 min [9,10].

Greater father involvement in child related activities decreases the women’s burden of childcare, which is related to better maternal mental health and lower stress [4,11,12]. The quality of this association may be explained by coparenting, which is the ability of parents to work together and share responsibilities in their parental roles [13], with positive coparenting linked to overall positive parental practices and maternal wellbeing [14,15]. However, the explicative role of coparenting on this association has been barely tapped by research. The current study aims to expand this corpus of research by examining whether greater father involvement with childcare is related to lower maternal stress through coparenting.

### 1.1. Maternal Stress

Overall, stress may be defined as an adverse psychological and behavioral response resulting from the imbalance between perceived demands and the available resources to manage them [16]. Parental stress is specifically related to the performance of parental tasks, encompassing a persistent perception that parental demands exceed available resources, including: feeling of being overwhelmed with parental tasks; feeling trapped; and a strained relationship with the child [17,18].

In theory, both parents may experience parental stress, but data showed us that it is typically faced by women [3,8,19,20,21,22]. This possibly happens because, despite the greater involvement of women in paid work, they remain in charge of most of domestic and childcare activities when compared to men [3,4,23]. On average women spend between two and ten times more time in these activities than men [24,25,26]. This data suggests that despite men’s greater involvement in domestic and familiar tasks, it is still not enough to decrease women’s burden [3,6]. The same happens in Portugal with women reporting a weekly average of 24.5 h in activities related to domestic activities, when men reported 8.5 h [27].

The greater vulnerability of mothers to parental stress may also be related to the type of child’s daily tasks in which they are involved. They tend to be more involved with emotionally demanding childcare activities; whereas fathers tend to do more fun things with the child, especially when it comes to the proportion of time spent on leisure activities rather than routine chores [3,28,29]. Moreover, managing child’s emotional demands is also critical due to their unpredictability, which negatively impacts women’s perception of free time, accounting for the greater perception of stress. On the one hand, women have less time to engage in hobbies or leisure activities, apart from the child and house. On the other hand, the lack of free time limits their ability to recover from the overwhelm of continuous childcare [30,31]. The failure to control their time and the continuous experience of fragmented time, which typically happens in parenting during the child’s first years, are highlighted as critical aspects for stress [3,8]. In line with this, maternal stress tends to be higher when the child is younger, due to its greater dependence from parents, involving more fragmented time and demanding more care time, particularly from women [3,8,32]. Indeed, studies have been uncovering how mothers tend to simultaneously manage a primary (domestic) activity while also doing childcare (e.g., cooking while entertaining the child), which is much less frequent for fathers [3,30].

The negative consequences of maternal stress on the quality of parental behaviors and child socioemotional outcomes are well described in the literature [33,34,35,36,37,38,39], and are being addressed as a major public health issue by the World Health Organization [40,41]. However, maternal stress is influenced by a set of ecological and interpersonal aspects. For instance, greater economic and social resources decrease the perception of maternal stress [3,42,43]. In what concerns parents’ relationship, father involvement in a child’s caregiving decreases maternal perception of stress [3,12], as detailed next.

### 1.2. Father Involvement, Coparenting and Maternal Stress

Father involvement is a broad concept encompassing the multiple ways in which fathers participate in a child’s life [4,44]. It may include direct interactions with the child in diverse activities, such as caregiving (e.g., feeding, bathing, dressing), playing with the child, teaching new things, or supervising activities and behaviors, as well as indirect care, in which fathers contribute to childcare and wellbeing without being directly involved with the child (e.g., preparing meals or planning routines) [12,45,46,47,48]. However, most of the studies focused on paternal engagement in activities related to fostering child development, such as reading a book or play games, leaving unexplored the extent to which fathers are involved with other activities [4,11,12,23,49]. This is critical given that the different ways in which fathers may be involved with their child—e.g., leisure/play vs. caregiving activities—may differently account for the perception of maternal stress [11,12,50,51].

It is discussed that father’s greater involvement in activities related to direct—i.e., performative tasks—and indirect care—i.e., managerial tasks—may be more helpful to decrease maternal stress than their involvement in child’s play and leisure activities. This distinction relies on the father’s ability to immediately support mothers in managing the multiple tasks related to child and house chores [11,12,48]. Nevertheless, the directions between father involvement and maternal stress are not yet clearly stated. For instance, the perception of greater father’s involvement in childcare tasks is related to lower maternal stress [11,52,53,54]; however, the absence of this association was also found [51,55].

These contradictory findings may rely on indirect effects which have remained mostly underexplored. Among the mediators, the quality of the mother–father relationship or romantic relationship were the most often examined, e.g., [52,54]. However, the ability of parents to share responsibilities and decisions related to parental tasks and childcare—i.e., coparenting [13,56,57]—may play an important role to better explain the association between father involvement and maternal stress. Coparenting may be positive or negative. It is positive when both parents are able to support each other concerning their parental roles and working together to share decisions and activities concerning childcare, i.e., cooperative coparenting [13,57]. Negative coparenting happens when parents are not able to set shared goals concerning their commitment on parental roles and attempt to weaken their partner’s position—i.e., conflict coparenting—or when one of the parents makes an alliance with the child against the other one—i.e., triangulation coparenting.

The way parents communicate and set their parental roles may allow for better understanding of the interplay between father involvement and maternal stress. Greater father involvement has been related to the positive form of coparenting [45,52]. Additionally, cooperative coparenting makes it easier for mothers to manage parental tasks, decreasing their perception of distress [3,11,55,57].

The current study adds to the current literature by examining the links between father involvement, maternal stress, and coparenting. Importantly, it will focus on mother’s perceptions about father involvement in caregiving activities to better understand its contribution to maternal stress [12,51,52]. Moreover, the literature related to maternal stress has been mainly concentrated in American families from impoverished backgrounds [3,11,52], leaving unexplored how father involvement contributes to maternal stress in dual-earner, middle-class families, which correspond to most of the families in European countries. The current study aims to overcome some of these limitations, to better understand how father involvement in childcare contributes to decreased maternal stress. It will accomplish it by examining the direct effects of father involvement and the mediating role of coparenting on maternal stress on a sample of Portuguese working mothers.

## 2. Materials and Methods

### 2.1. Participants

Mothers (*n* = 254) of children aged 2–6 years old (*M* = 4.87; *SD* = 1.68; 54.2% boys) were recruited in public and private schools. Only working and married/cohabiting mothers were included in data analysis. On average, mothers were 37.67 years old (*SD* = 5.31; range: 21–51 years old), with 68% of mothers reporting to hold a college degree. Almost half of the mothers (46.3%) reported working between 35–40 h/week, which corresponds to the Portuguese legal work journey. However, 33.8% of the mothers reported working more than 40 h/week, and 20% reported working less than 35 h/week.

### 2.2. Procedures

The study was presented in schools/kindergartens and parents were invited to participate. Those who agreed received a letter explaining the study, a consent form, and a set of questionnaires assessing sociodemographic characteristics of the family, parental involvement, maternal stress, and coparenting. Some of the schools requested a link to the questionnaires, rather than the paper format for data collection. Thus, a link to the survey was created on Qualtrics and disseminated by schools among their mailing lists. This link was also shared by the research team on social media platforms. In the current study 21% of the data was collected online (after eliminating questionnaires due to missing data). Previously, the study was approved by the Ethics Committee of the university and the board of directors of the schools.

Mothers individually reported sociodemographic data about themselves, their children, and their family, as well as a set of measurements assessing their perception of father involvement, maternal stress, and coparenting. All these measures, except the sociodemographic questionnaire, were presented at random.

### 2.3. Measures

Sociodemographic questionnaire. A set of objective questions was developed by the research team, assessing parents’ sociodemographic status, e.g., age, education, marital status, and family characteristics, e.g., place of residence, number of children, child’s sex, age. Aspects of their professional lives were also assessed, e.g., working status, number of daily working hours. Fathers reported family’s sociodemographic characteristics.

Parental Involvement Scale: Caretaking and Socialization Activities [58]. It is a 26 items self-reported questionnaire examining father’s perception about his involvement, in relation to the mother, with child related activities. It is answered in a 5-point scale (from “always the mother” to “always the father”), in which higher scores correspond to greater father involvement across five dimensions of care: direct and indirect care, education/teaching, play, and leisure. In this study we only included the subscale of direct and indirect care, due to our main interest in examining father involvement in childcare activities. Direct Care assesses the direct involvement in child’s daily activities of care, such as feeding meals and putting to bed (e.g., who bathes the child); Indirect Care examines aspects related to decision making and responsibility for activities related to child’s wellbeing without direct interaction, such as scheduling doctor’s appointments and preparing meals (e.g., who buys the child’s clothes). Acceptable internal consistency reliabilities were obtained for both Direct (0.67) and Indirect care (0.65).

Parenting Stress Scale [Portuguese version; [59]]. It is a 17-items scale that evaluated parental stress in four main dimensions: parental worries (e.g., to take care of my child is my main stressor); satisfaction with parental role (e.g., I like to spend time with my child); fear (e.g., sometimes, I wonder if I do enough for my child); and lack of control (e.g., to have child(ren) means unpredictability and lack of control in my life). Mothers answered in a five-point Likert scale, in which higher scores correspond to higher stress. The dimension of satisfaction is reverse coded, so higher punctuation corresponded to lower satisfaction. Good internal consistency values were obtained from all dimensions, varying from 0.77 to parental worries; 0.71 to satisfaction; 0.76 to fear; and 0.69 to lack of control.

Coparenting Scale [Portuguese version; [60]]. It evaluates the coparenting quality through 14 items, in which each partner evaluated the other one, concerning parental tasks, corresponding to the three types of coparenting: cooperative (e.g., my partner talks with me about our child); conflict (e.g., my partner argues with me because of our child); triangulation (e.g., my partner talks badly about me in front of our child). Mothers answered in a five-point Likert scale, in which higher values correspond to the type of coparenting. Good internal reliability values were obtained for the dimension of cooperation (0.84) and conflict (0.77). The dimension of triangulation obtained low levels (0.54) and was not included in the analysis.

### 2.4. Data Analysis

Data was analyzed using SPSS (Version 28; IBM SPSS Statistics, Inc., Chicago, IL, USA). Firstly, descriptive statistics were performed on all sociodemographic and study variables. Pearson correlations examined bivariate associations between sociodemographic variables, the dimension of direct & indirect care of father involvement, dimensions of maternal stress, and the dimensions of cooperative and conflict coparenting. Maternal age, education, and number of hours at work were not correlated with study variables (i.e., father involvement in direct & indirect care; coparenting and maternal stress) and were not included in further analysis. Only variables significantly correlated were included in the mediation analyses.

To test whether the effects of father involvement on mothers’ stress were mediated by coparenting simple mediation was conducted using PROCESS macro for SPS [Model 4; [61]] and four models were tested. Mediation analysis is conducted to understand the mechanisms through which one independent variable will influence the dependent variable. Therefore, there are two possible ways in which the independent variable influences the dependent variable: directly or indirectly through the mediator [61]. In the first model, involvement in direct care was the independent variable, cooperative coparenting was the mediator, and the satisfaction dimension of maternal stress was the dependent variable. In the second model, involvement in direct care was the independent variable, conflict coparenting was the mediator, and the satisfaction dimension of maternal stress was the dependent variable. In the third model, father involvement in indirect care was the independent variable, cooperative coparenting was the mediator, and lack of control was the dependent variable. Finally, in the fourth model, father involvement in indirect care was the independent variable, conflict coparenting was the mediator, and lack of control was the dependent variable. Bootstrapping is a statistical technique used to estimate confidence intervals for mediation effects, helping to determine whether the mediation effect is statistically significant [61]. Indirect effects were subjected to bootstrap analyses (bias-corrected) with 5000 samples and a 95% percentile confidence interval estimate (CI). The effect was considered significant when the CI does not include zero [61]. R-squared was used to estimate the effect size of each statistically significant model [62].

## 3. Results

### 3.1. Preliminary Analysis

Descriptive statistics and correlations among the study variables are displayed in Table 1. Maternal perception of existing father involvement in direct care was related with lower mothers’ satisfaction on parental role, greater cooperative coparenting, and less conflict coparenting. Maternal perception of existing father involvement in indirect care was negatively related to lack of control on parental roles and conflictual coparenting. Greater involvement in indirect care was also related to positive coparenting. No relation was found between sociodemographic variables (mother’s age, education, and working hours) and the studied variables.

### 3.2. Mediation Models

Results show the association between father involvement in (in)direct care and mother’s (lack of) satisfaction and lack of control, mediated by cooperative and conflict coparenting (Table 2). Direct paths from the first model revealed a positive association between father involvement in direct care and cooperation, but the association between involvement in direct care and mother’s (lack of) satisfaction was also positive. However, this association becomes negative in the presence of cooperative coparenting; this suggests that, when mothers perceive positive forms of coparenting, greater father involvement in direct care account for more satisfaction on maternal role, with large effect size. Cooperative coparenting plays a partial mediating role, significantly predicting mother’s satisfaction (*F*(2; 234) = 17.38, *p* < 0.001), explaining 13% of its variance.

This pattern is repeated in the second model, while examining the mediating role of conflict coparenting in the association between father involvement in direct care and maternal (lack of) satisfaction. Direct paths showed a positive association between direct care and mother’s (lack of) satisfaction, but when conflict coparenting was added as a mediator this relation changed direction. Results suggest that when fathers are less involved in direct care, mothers perceive higher conflict in coparenting and their satisfaction on maternal role is reduced, with medium effect size. Findings revealed that conflict coparenting partially explained this association, with the model predicting 15% of its variance (*F*(2; 236) = 20.78, *p* < 0.001).

The third model analyzed the mediating role of cooperative coparenting on the association between father involvement in indirect care and mother’s lack of control. Findings revealed a nonsignificant direct path between father involvement in indirect care and mother’s perception of lack of control. However, the indirect path mediated by cooperative coparenting showed significant associations; this indicates that, when mothers perceived positive forms of coparenting, greater father involvement in indirect care explains reduced maternal feelings of lack of control, with large effect size. Cooperative coparenting plays a complete mediating role, significantly predicting mother’s lack of control (*F*(2; 236) = 18.14, *p* < 0.001), and explaining 13% of its variance.

Finally, the fourth model examined the mediating role of conflict coparenting in the association between father involvement in indirect care and mother’s lack of control. Findings again revealed a nonsignificant direct path between father involvement in indirect care and maternal perception of lack of control. Equivalent to the third model, the indirect path in the presence of conflict coparenting showed significant associations; this indicates that, when mothers perceived higher conflict in coparenting, reduced father involvement in indirect care account for greater mother’s feeling of lack of control, with large effect size. Results depicted that the association between father involvement in indirect care and mother’s lack of control fully happens through conflict coparenting, predicting 15% of its variance (*F*(2; 239) = 21.00, *p* < 0.001).

Figure 1 illustrates an integration of main findings from the four models.

## 4. Discussion

The present study examined the mediating role of coparenting in the interplay between father involvement in childcare and maternal stress. Findings uncovered the critical role of coparenting to differently mediate the association between father involvement in childcare and maternal stress.

Specifically, the results indicated the critical role of cooperative coparenting in the association between father involvement both in direct and indirect care to decrease the perception of maternal stress. This result suggests that greater father involvement in both dimensions of care per se does not mean decreased maternal stress, particularly in the dimensions of satisfaction with the maternal role and lack of control. The analysis of direct paths revealed that greater father involvement in child’s caregiving tasks is related with lower mother’s satisfaction in their maternal role. This may rely on the social construction of motherhood and women’s role in Portuguese society, in which it is expected that mothers are able to conciliate both maternal and professional demands [63]. Mothers who perceived child’s father as more involved in activities related to child’s caregiving may evaluate themselves as failing on their role as mothers by not fully managing child’s caregiving [20,64]. Women who perceived fathers as more engaged in their parental tasks may evaluate themselves as failing in their expected roles, given the social construction and validation of women’s ability to multitask: namely, to perform well in professional, maternal, and domestic activities. This may contribute to lower parental satisfaction, as well as the perception of lack of control on parental activities [3,65]. Other explanations to current findings may rely on typical mothers’ “managerial role” supervising fathers’ involvement with childcare activities. By doing that mothers remain mentally involved with childcare tasks, which is not enough to decrease the burden of care [3,66].

Interestingly, this association changes in the presence of cooperative coparenting, suggesting the importance of parents’ ability to jointly set shared parental goals and activities [13,50,57]. Moreover, it is discussed that (cooperative) coparenting can be considered an aspect of social support [64], which may explain its positive influence in decreasing maternal stress. When that happened there was a decrease of maternal stress both on maternal satisfaction and lack of control. This finding highlights the critical role of cooperative coparenting in decreasing maternal stress.

In opposition, the presence of conflictual coparenting increases maternal perception of stress both in the dimensions of satisfaction and control. It is also discussed that greater conflict on parental roles, namely by the inability to be equally committed to parental roles, contributes to greater maternal stress, as it was observed in our study. This result underlies the previous finding by pointing out the critical role of coparenting in parents’ mental health and wellbeing, e.g., [14,15]. When mothers perceived their relationship with their partner as involving conflict and tension, a negative spillover effect is observed, increasing maternal stress, which may strain the quality of the relationship with the child, e.g., [34,35,39]. In these cases, conflictual coparenting does not help to decrease the feeling of being overwhelmed with parental tasks [17,18], and can even lead to mother’s avoidance of father’s participation [67]. Additionally, recent findings discussed how fathers’ greater involvement in childcare is associated with less conflict coparenting [67], which this study also found.

It is important to address that reports are driven from mother’s perspective, which may influence results. On one hand, mother’s experience greater parenting stress when available resources do not match with their expectations about their needs to meet demands of parenting [68]. Therefore, mother’s report of parenting stress will depend on their expectations regarding father involvement [11]. On the other hand, mothers tend to report lower levels of father involvement than fathers [69,70], especially when there is higher parenting conflict [69]; so, it becomes difficult to know which one is more reliable. Additionally, since we cannot draw conclusions about causal directions, it is impossible to know if father involvement causes mother stress or the other way around. Because father involvement often relies on mothers’ encouragement of father’s involvement, e.g., [66,71], it may happen that mothers experiencing greater parenting stress feel an additional burden to facilitate father–child interaction, leading to lower father involvement [11].

### Limitations and Future Directions

Despite the relevance of these findings, some limitations must be addressed. Firstly, the reliance on self-reported measures, which may be affected by social desirability effects. Hence, the use of a triangulation of measures is critical, such as diary studies or observations. Secondly, the cross-sectionality of the data limited the understanding of how these variables may evolve over time. This is particularly relevant given that maternal stress tends to be higher at child’s early years, decreasing over time, e.g., [65]. Third, despite the effort to obtain a socially diverse sample, most of the participants were middle-class mothers with high education and similar working hours, limiting the understanding of the influence of demographic aspects, such as income, number of working hours, or education, which should be addressed by future research. Also, it would be beneficial to look deeper into cooperation and conflict constituents, making clearer the type and nature of coparenting participation in this association. Despite mothers’ perspective being the focus of this study, future research on this topic should consider dyadic interactions, looking at both mothers and fathers’ reports. Finally, findings do not allow us to understand how maternal stress affects parental practices. Future studies may want to examine how the interplay between coparenting and maternal stress contributes to parental practices and child socioemotional outcomes, namely through observational methods and longitudinal designs.

## 5. Conclusions

The present work focuses on the relationships between maternal stress, father involvement, and coparenting. This is an important topic at both social and scientific levels due to the changes in family dynamics and parenting roles of the last years. Our findings shed light on how despite the increasing of father involvement in childcare [1,2], particularly in dual-earners middle class families, mothers are still overwhelmed by the need to balance both work and family/childcare tasks [3]. Findings of the current study are relevant at theoretical, empirical, and clinical points of view. Findings uncovered how father involvement may differently account for maternal stress through coparenting, suggesting that father involvement in childcare is not enough to decrease maternal stress. This result may have implications for couple and parenting interventions. For instance, instead of interventions focusing on behavioral changes in interactions, they may want to target gendered dynamics of parenting as antecedents of maternal stress, e.g., [65]. Naturally, these changes should be supported at the social level with policies supporting shared and equitable parenting and less stressor environments for parenthood [3,72]. That being said, we believe that the obtained findings are relevant by providing evidence of dimensions of parental relationship that contribute to maternal stress, highlighting the critical role of coparenting on it.

## Figures and Tables

**Figure 1 ijerph-20-05457-f001:**
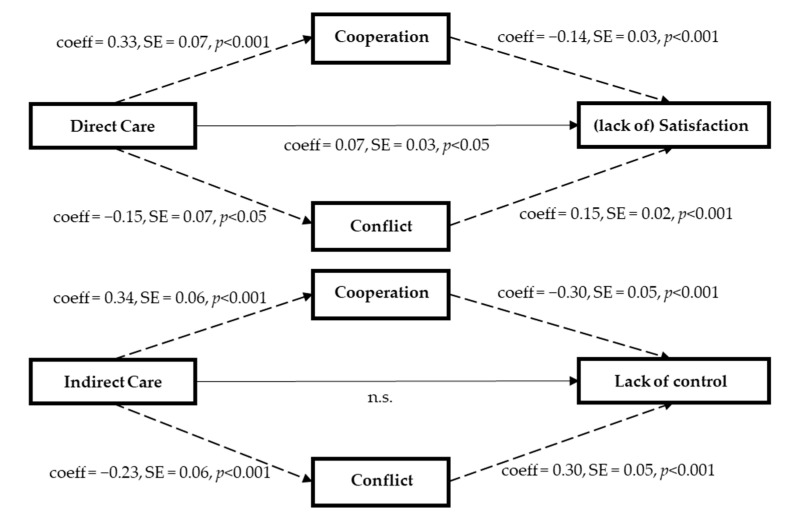
Understanding Regression Coefficients for the relationship between Father Involvement, Coparenting, and Mother’s Stress. *n* = 254; 5000 bootstrap sample. Bootstrap bias corrected *p* values. Only significant indirect effects were reported. coeff = nonstandardized estimate; *SE* = standard error. Solid arrows indicate direct effects and dotted arrows indicate indirect effects.

**Table 1 ijerph-20-05457-t001:** Pearson Correlations and Descriptive Statistics among Studied Variables (*n* = 254).

	*M* (*SD*); Range	1	2	3	4	5	6	7	8
1. Direct care	2.40 (0.60); 1–4	-	-						
2. Indirect care	2.30 (0.48); 1–4	0.48 **	-						
3. Maternal worries	2.19 (0.81); 1–4	0.012	−0.09	-					
4. (lack of) Satisfaction	3.34 (0.18); 1–4	0.12 *	−0.08	0.26 **	-				
5. Fear	3.74 (1.02); 1–5	−0.04	−0.07	0.35 **	0.06	-			
6. Lack of control	1.45 (0.53); 1–4	−0.03	−0.12 *	0.53 **	0.57 **	0.27 **	-		
7. Cooperation	4.34 (0.66); 1–5	0.30 **	0.34 **	−0.23 **	−0.23 **	−0.08	−0.34 **	-	
8. Conflict	2.23 (0.51); 1–4	−0.14 *	−0.16 *	0.35 **	0.33 **	0.21 **	−0.38 **	−0.49 **	-

* *p* < 0.05; ** *p* < 0.01.

**Table 2 ijerph-20-05457-t002:** Bootstrap Test for Indirect Effects from Father Involvement in (In)Direct Care to Mother’s (lack of) Satisfaction and Uncontrol.

		Bootstrapping		
		Bias Corrected: 95% CI for Mean Indirect Effect		
Effect	*B*	*SE*	Lower	Upper
Direct Care → Cooperation → (lack of) Satisfaction	−0.10	0.03	−0.17	−0.04
Direct Care → Conflict → (lack of) Satisfaction	−0.05	0.03	−0.11	−0.01
Indirect Care → Cooperation → Uncontrol	−0.13	0.04	−0.21	−0.05
Indirect Care → Conflict → Uncontrol	−0.09	0.03	−0.15	−0.03

Note: CI = Confidence Interval.

## Data Availability

Data are available under request to corresponding author.

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
