# Peer review of "Father Involvement and Maternal Stress: The Mediating Role of Coparenting"

_ijerph, 2023, doi:10.3390/ijerph20085457_

Round 1

Reviewer 1 Report

I compliment with the authors for their well-conducted and written research on father importance in childcare. However, the results should be discussed stressing more that the study involved only mothers who were asked to report on fathers' parenting. How this may have influenced the results? Please, discuss this issue more in depth.

Author Response

We thank the Reviewer by the opportunity to clarify our methodological options and its implications to the state of the art. We now better discussed how mothers’ reports may have accounted to results (please see lines 342-353).

Reviewer 2 Report

This paper is interesting and makes important points regarding caregiving and maternal stress. The paper would benefit from the following:

·      An intensive English edit

·      A clearer explanation for the choices made in the analysis. For example, why the focus on those two aspects of maternal stress?

·      A clear explanation of the models used (and not just the statistical package).

Author Response

  1. An intensive English edit.

            A full revision of the manuscript was made by a professional reviewer.

  1. A clearer explanation for the choices made in the analysis. For example, why the focus on those two aspects of maternal stress?

We thank R#2 for the opportunity to clarify our analysis plan. We choose to include only variables significantly correlated in the mediation analyses (as detailed in Table 1). This information is now detailed in the manuscript (please see line 214).

  1. A clear explanation of the models used (and not just the statistical package).

Again, we thank the opportunity to clarify our analysis. For this purpose, we added the following explanation (lines 216-222: “To test whether the effects of father involvement on mothers’ stress were mediated by coparenting simple mediation was conducted using PROCESS macro for SPSS [Model 4; 61] and four models were tested. A mediation model can be defined as a causal model, in which one independent variable will influence the dependent variable through one only and intervenient variable (mediator). Therefore, there are two possible ways in which the independent variable influences the dependent variable: directly or indirectly through the mediator [61].”.

Reviewer 3 Report

Dear Authors,

the topic is very interesting and practical. The literature review is well done. The research process is not objectionable. The research was well thought out, well designed. The results are rather predictable. They did not come as a surprise. This is socially relevant. The research has a utilitarian dimension. They indicate how much more there is to do in this regard. The text has an important message. The only caveat is that the conclusions are a bit laconic. The discussion of the results is interesting. I recommend the text for publication.

Author Response

  1. The only caveat [of the manuscript] is that the conclusions are a bit laconic.

We thank the reviewer for the opportunity to improve our conclusions; now we better framed the study's conclusions (please see lines 375-380).

Reviewer 4 Report

Dear authors, your research is very interesting and and provides useful insights in father involvement in childcare.

Please consider a few comments as listed below:

line 144, pls explain "Mage"

explain why you applied mediation models and bootstrapping for your data analysis.

Finally, it would benefit your research if you could further analyse cooperation into its constituents, thus making it more clear the type and nature of collaboration.

Author Response

  1. Line 144, pls explain "Mage".

To avoid misunderstandings, we now replaced Mage by M.

  1. Explain why you applied mediation models and bootstrapping for your data analysis.

            We thank the reviewer for the opportunity to clarify our analytic plan. We now detailed the goal of mediation models in the Data Analysis section (please see lines 218-222) and bootstrapping (lines 230-232).

  1. Finally, it would benefit your research if you could further analyse cooperation into its constituents, thus making it more clear the type and nature of collaboration

            Thank you for your feedback, which was taken into great consideration. We can recognize it as a limitation; therefore, it was included as a suggestion for future research (please see lines 364-366).

Round 2

Reviewer 2 Report

I thank the authors for their edits. I remain confused about how the models are estimated. Are these OLS models with interaction effects? I'm also not sure why the authors call these causal models, as they are not.

I'm also confused by the presentation of the results: effect sizes are not discussed and there is an emphasis on the R-squared that I do not think is warranted in this empirical context.

Again, I think this is an interesting paper but the methods and therefore results remain a bit obscured to me.

Author Response

I thank the authors for their edits. I remain confused about how the models are estimated. Are these OLS models with interaction effects? I'm also not sure why the authors call these causal models, as they are not. 

We thank the reviewer for the opportunity to clarify our explanation of the model. The definition of a mediation model was changed, removing the word “causality” (please see lines 218-220). We also address that nor the interpretation of results, neither their discussion was made in a perspective of causality. We also clarify, that according to Sewal Wright, mediation analysis “is not a method for discovering causes, but a method applied to a causal model already formulated on the basis of knowledge and theoretical considerations” and “the method of path coefficients is not intended to accomplish the impossible task of deducing causal relations from the values of the correlation’s coefficients. It is intended to combine the quantitative information given by the correlations with such qualitative information as may be at hand on causal relations to give a quantitative interpretation” (Wright, 1934, p. 193). 

I'm also confused by the presentation of the results: effect sizes are not discussed and there is an emphasis on the R-squared that I do not think is warranted in this empirical context. 

Thank you for your feedback, which was taken into significant consideration. As described in the literature (Fairchild et al., 2009), we used R-squared as a measure of the effect size on global mediation models (please see lines 236-238). Nevertheless, as requested, we added interpretations of individual mediation interactions - another measure of effect size - in the Results section.